# Exploring the Sensitivity of LLMs' Decision-Making Capabilities: Insights from Prompt Variation and Hyperparameters

**Manikanta Loya***
manikanl@uci.edu

**Divya Anand Sinha***
dasinha@uci.edu

**Richard Futrell**
rfutrell@uci.edu

University of California, Irvine

## Abstract

The advancement of Large Language Models (LLMs) has led to their widespread use across a broad spectrum of tasks, including decision-making. Prior studies have compared the decision-making abilities of LLMs with those of humans from a psychological perspective. However, these studies have not always properly accounted for the sensitivity of LLMs' behavior to hyperparameters and variations in the prompt. In this study, we examine LLMs' performance on the Horizon decision-making task studied by Binz and Schulz (2023), analyzing how LLMs respond to variations in prompts and hyperparameters. By experimenting on three OpenAI language models possessing different capabilities, we observe that the decision-making abilities fluctuate based on the input prompts and temperature settings. Contrary to previous findings, language models display a human-like exploration–exploitation tradeoff after simple adjustments to the prompt. [1]

## 1 Introduction

The recent success of large language models (LLMs) at a variety of tasks has led to curiosity about their cognitive abilities and characteristics. As LLMs are increasingly integrated in daily life both as conversation partners and economic decision-makers (Munir et al., 2023; Chaturvedi et al., 2023; Yang et al., 2023), such studies are necessary for understanding the limits and characteristics of such agents. An understanding of LLMs at a psychological level may also provide strategies for improved prompting and training. To this end, a number of researchers have recently adopted methods from cognitive psychology and behavioral economics to evaluate language models in the same way that humans have been evaluated (e.g. Linzen et al., 2016; Miotto et al., 2022; Phelps and Russell, 2023, among many others).

However, such work has not always paid due attention to the fact that LLM responses can be highly variable and sensitive to the details of the prompt used and to hyperparameters such as temperature. Limited interactions with LLMs—such as interactions using only one prompt—can be misleading (Bowman, 2023). In this work, we follow up on the behavioral experiments conducted by Binz and Schulz (2023), who studied LLMs' decision making using analogues of a number of well-known human experimental paradigms, finding strong divergences from human behavior. However, the experiments in the previous work used only one prompt per task, and did not study the effects of hyperparameters. We adopt the same task as the previous work, but systematically vary prompts and temperature.

Our aims are both substantive—we seek to find whether, with basic changes to the prompt, models show human-like behavior in these decision making tasks—and methodological: we wish to emphasize that psychological LLM research must consider variability as a function of prompt and hyperparameters.

## 2 Background and Related Work

As mechanistic understanding and control of LLMs remains complex, researchers have increasingly adopted methods from human behavioral sciences for characterizing LLMs' behavior: in the same way that the human brain is largely a black box that must be probed using experimental methods and constructs, LLMs may be studied in the same way.

In addition to studies that have used the methods of cognitive psychology to understand LLMs' reasoning and grammatical abilities (e.g., Linzen et al., 2016; Futrell et al., 2019; Cai et al., 2023), researchers have increasingly adapted methods from psychometrics (Miotto et al., 2022; Bodroza et al., 2023; Abramski et al., 2023), which seek to characterize LLMs in terms of personality variables such

---

*Equal Contribution

[1]Code is available at the following github link.

You are going to a casino that owns two slot machines. You earn money each time you play on one of these machines.

You have received the following amount of dollars when playing in the past:
- Machine F delivered 51 dollars.
- Machine J delivered 39 dollars.
- Machine J delivered 40 dollars.
- Machine J delivered 26 dollars.

Your goal is to maximize the sum of received dollars within six additional rounds.

Q: Which machine do you choose?
A: Machine

Figure 1: Original Horizon 6 task prompt (Binz and Schulz, 2023).

as agreeableness and conscientiousness, and methods from behavioral economics (Cartwright, 2018; Phelps and Russell, 2023; Horton, 2023), which characterize LLMs' decision-making in terms of preferences for risk and reward.

Prior research on prompting techniques (Wei et al., 2022; Wang et al., 2023) has shown that subtle modifications in input prompts can lead to varied outcomes in reasoning tasks (Cobbe et al., 2021). Srivastava et al. (2022) revealed that Large Language Models (LLMs) are notably susceptible to the precise wording of natural language questions, especially when presented in a multiple-choice setting. In a recent study, Ouyang et al. (2023) emphasized the influence of temperature adjustments on LLM's performance in code generation tasks. Unlike previous studies, our research delves into the sensitivity of LLMs concerning economic decision-making abilities.

Our work is a focused followup on Binz and Schulz (2023), investigating the sensitivity of one of their results to changes in prompt and hyperparameters. Binz and Schulz (2023) evaluated on decision-making, information search, deliberation, and causal reasoning in text-davinci-002 (Brown et al., 2020) by presenting it with prompts such as the one shown in Figure 1. We follow up on the tasks from the information search area, instantiated in the Horizon task, described in the following section. In this task, humans show a characteristic trade-off of exploration and exploitation (Wilson et al., 2014), favoring exploration in early trials and exploitation later, whereas Binz and Schulz (2023) find that LLMs do not.

The results of Binz and Schulz (2023), however, are based on single prompt and setting, limiting the generality of their results. Furthermore, observing the Horizon task prompt (and the others used throughout the paper), it does not follow what

are now regarded as best practices for such tasks, for example the use of Chain-of-Thought (CoT) prompting (Wei et al., 2022)—the original prompt forces the LLM to choose a machine in the next token generated, without deliberation. Below, we investigate the behavior of LLMs on this task under systematic variations of temperature and prompt.

## 3 Horizon Task Experiments

The Horizon Task as shown in Binz and Schulz (2023) is a special case of the Multi-Armed bandit (MAB) setting. MAB problems (Sutton and Barto, 2018, Ch. 2) are one of the common problems in the area of Reinforcement Learning. This game involves an agent interacting with a slot machine possessing $k$ arms. Each arm the agent pulls has a reward associated with it defined by an underlying probability distribution. This game is played over multiple episodes with the goal of maximizing the accrued rewards.

One approach involves persistently selecting the arm that has delivered the maximum amount of rewards in the past. An alternative strategy involves thorough exploration of all arms to discern their respective underlying probability distributions, followed by the selection of the arm with the highest potential for reward. While this is feasible, every turn spent in discerning the underlying distribution, diverts from the primary goal of reward maximization. The former and latter strategies are known as **exploitation** and **exploration** respectively, and the exploration–exploitation dilemma is a fundamental concept in decision-making that arises in many domains.

To explore the extent to which humans use these strategies, Wilson et al. (2014) reports experiments where participants were asked to play the Horizon Task. This task consists of a set of two-armed bandit problems, where participants are presented with two options, each associated with noisy rewards. The task comprises either five or ten trials, and in each trial, participants must select one option, receiving corresponding reward feedback. In the initial four trials of the task, participants have only one option and are provided with the corresponding reward feedback. These forced-choice trials create two distinct information conditions: "unequal information" and "equal information." In the unequal information condition, one option is played three times, while the other option is played only once. In the equal information condition, both options are

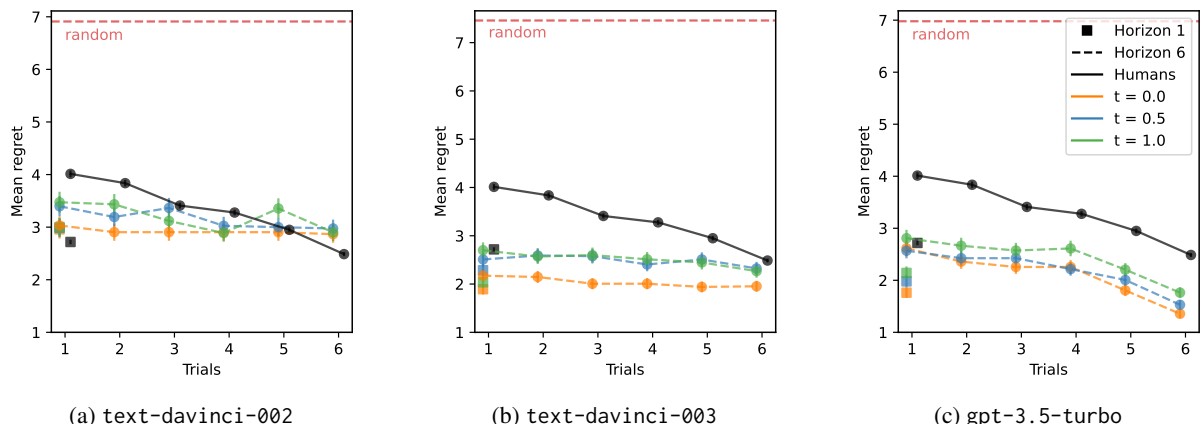

(a) text-davinci-002      (b) text-davinci-003      (c) gpt-3.5-turbo

Figure 2: Mean regret obtained in the Horizon (multi-trial multi-armed bandit) task by humans and LLMs with varying temperature, using the prompt from . The solid black line indicates human performance; others are LLMs. Error bars show the standard error of the mean.

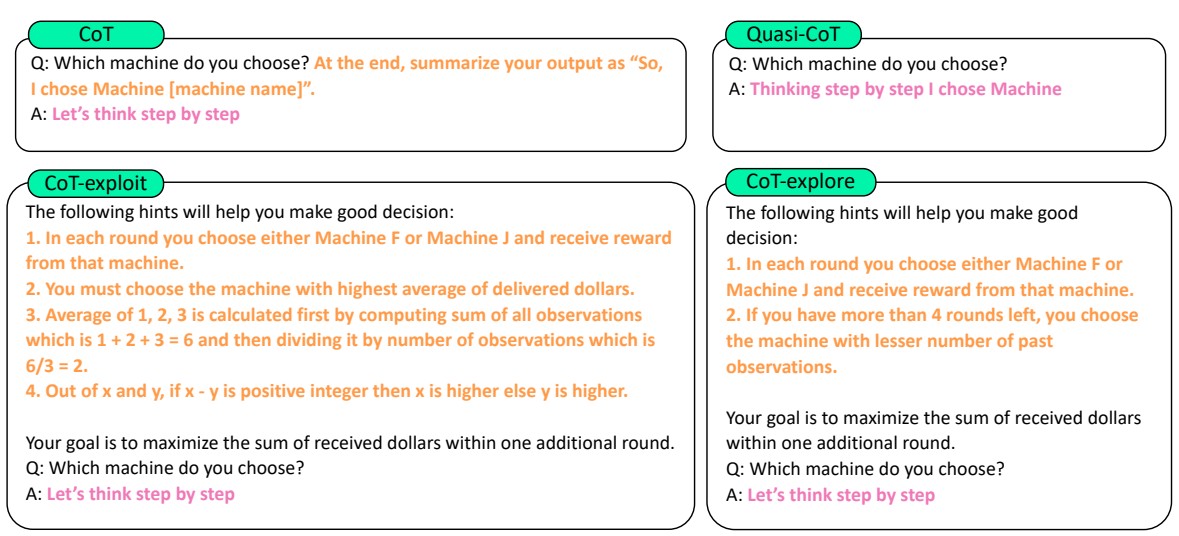

Figure 3: Modifications in prompt for the Horizon task. Horizon 1 prompt is shown. In case of CoT, CoT-Exploit & CoT-Explore we explicit ask the model to summarize its choice at the end by appending the entire prompt with "Answer the following question and summarize your choice at the end as 'Machine:[machine_name]'." at the beginning.

played twice. The five-trial setting is denoted **Horizon 1**, indicating that participants make decisions only once, while the ten-trial setting is referred to as **Horizon 6**, as participants make decisions over six rounds.

Binz and Schulz (2023) applied this experimental design to language models using the prompt in Figure 1, and we follow their experimental setup exactly except for variations to the prompt and hyperparameters. The performance of LLMs is assessed by measuring the mean regret across multiple runs. The regret is defined as the difference between the optimal reward, which corresponds to the machine with the higher reward, and the actual reward obtained from the selection process. Hu-

man behavior favors exploitation in Horizon 1, but a gradual shift from exploration to exploitation in Horizon 6.

### 3.1 Varying Temperature

The impact of various temperature settings (`0.0,0.5,1.0`) on all three OpenAI models[2] tested is illustrated in Figure 2. It is clear that the behavior of each model, as indicated by the mean regret line, differs according to the temperature. For Horizon 1, the lowest regret is obtained for temperature zero across all three models. Further, unlike text-davinci-002 and as shown in Binz and Schulz (2023), the mean regret is

[2]https://platform.openai.com/docs/models/overview

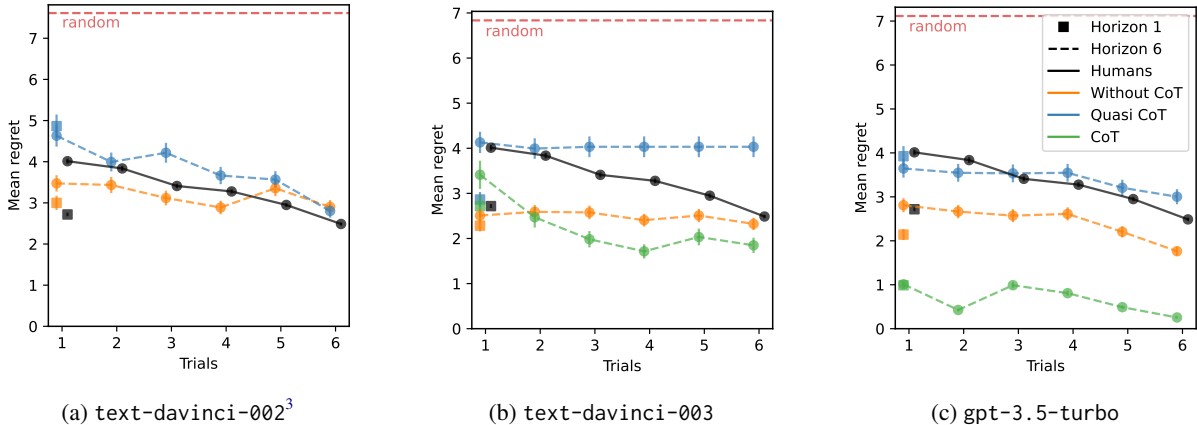

(a) `text-davinci-002`[3]  (b) `text-davinci-003`  (c) `gpt-3.5-turbo`

Figure 4: Mean regret obtained by humans and LLMs on the Horizon task, varying prompt. 'Quasi-CoT' means a prompt of the form 'Thinking step-by-step, I choose Machine . . . ' which does not enable true chain-of-thought reasoning. The temperatures for GPT-2, GPT-3, and GPT-3.5 are 1.0, 0.5, and 1.0 respectively. These temperatures show the greatest learning effect (negative slope) in the Horizon 6 task.

lower than humans for both `text-davinci-003` and `gpt-3.5-turbo`. In the case of Horizon 6, there is a notable rise in the inital mean regret, suggesting that higher temperatures result in suboptimal decision-making. However, increasing temperature demonstrates a more pronounced learning effect, as evidenced by a greater negative slope.

### 3.2 Varying Prompt

To encourage deliberation during decision-making, we incorporate variations in the input prompt. Specifically, we explore two different variants of the Chain of Thought (CoT) prompting technique (Wei et al., 2022)—CoT and Quasi-CoT. In Figure 3, we illustrate the modifications made to the original prompt. The variant referred to as Quasi-CoT utilizes the prompt "Thinking step by step I choose Machine", which forces the machine to make a decision before fully processing its reasoning. On the other hand, the CoT variant makes a decision only after fully processing its reasoning. The Quasi-CoT condition allows us to disentangle the effects of true step-by-step reasoning in CoT from the effects of prompting the LLM to think carefully.

The alteration in the behavior of LLMs due to changes in the input prompt is depicted in Figure 4. Across all models, CoT demonstrates lower-regret compared to both Quasi-CoT and the original prompt, whereas Quasi-CoT performs worse than original prompt. Furthermore, even in

`text-davinci-002`, we find that altered prompts yield the human-like negative slope, indicating an exploration–exploitation trade-off.

### 3.3 CoT Prompting with Hints

To overcome the identified limitations in LLMs, such as their inaccuracies in computing averages (Razeghi et al., 2022; Imani et al., 2023) and suboptimal exploration capabilities, we introduce additional hints within the input prompt to guide the decision-making process. Specifically, we designed two prompts, namely CoT-Exploit and CoT-Explore, which aim to facilitate explicit exploitation and exploration. The hints associated with these prompts are shown in Figure 3.

In the CoT-Exploit prompt, we instruct the model to base its decisions on the average of observed experiences and equip it with the required mathematical calculations to make a decision. Likewise, in the CoT-Explore approach, we explicitly direct the model to select a machine with lower frequency among the observed experiences.

The performance of `gpt-3.5-turbo`, using various CoT prompting variants, is compared in Figure 5. As anticipated, CoT-Exploit outperforms CoT-Explore, displaying a consistent decrease in slope. However, CoT-Explore performs significantly worse than random decision-making. CoT-Explore primary concentrates on getting more information about each machine rather than overall rewards.

---

[3]Experiments with `text-davinci-002` using CoT prompt failed due to its inability to summarize its choice at the end.

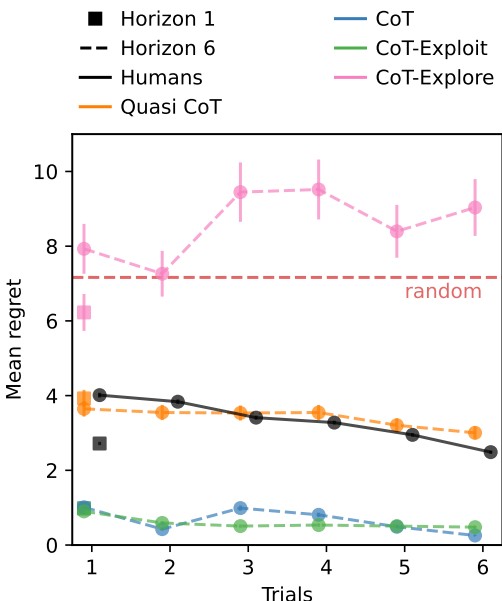

Figure 5: `gpt-3.5-turbo`'s behavior under different variants of CoT prompts at temperature 1.0.

### 3.4 Discussion

Through our experiments, we have discovered that the decision-making capabilities of LLMs are influenced by both the prompts used and the temperature settings, more so by the choice of prompt rather than the temperature. This highlights the importance of varying prompts to elicit the desired behavior from LLMs during decision-making tasks, and that studies which have used only one kind of prompt are potentially misleading.

Intriguingly, we observed that the model `gpt-3.5-turbo` with the Quasi-CoT prompt (Figure 5) exhibits the closest resemblance to human behavior. This prompt alerts the model to the need for reasoning, but does not give it the space to actually perform any reasoning. The similarity of the Quasi-CoT result to humans suggests that humans may also struggle to fully process the associated information and reasoning.

Furthermore, by providing hints to guide the decision-making process, we have observed that superhuman performance can be achieved, as demonstrated by the CoT-Exploit variant (Figure 5). This result suggests that language model behavior in these tasks is potentially highly controllable.

### 4 Conclusion

We have demonstrated that the psychological behavior of LLMs, as previously explored by Binz

and Schulz (2023), is highly sensitive to the way these LLMs are queried. The non-human-like behavior observed by Binz and Schulz (2023) vanishes under simple variations of prompt, and superhuman performance in terms of minimizing regret is easily achievable. Going forward, we urge careful consideration in the LLM psychology literature of the fact that model behavior can diverge under different settings.

### 5 Limitations

We have presented a focused extension of one of the studies from Binz and Schulz (2023), demonstrating sensitivity to prompt and hyperparameters which was overlooked in the previous work. However, our work is limited in that (1) we have only examined one of the tasks from Binz and Schulz (2023), (2) we have only presented a few variations of temperature and prompt, and (3) we have only experimented with some of the models available to us as of June 2023, selecting high-profile closed-source models over open-source models. Nevertheless, we believe that our overarching point that LLM psychology needs to take into account hyperparameters, prompts, and variability remains valid.

### Ethics Statement

This work involves psychological studies of LLMs in economic decision making contexts. If LLMs are really deployed as economic decision makers, then ethical issues could result from biases and limitations of the models. We urge caution in such applications.

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
