# OpenReview forum: "Exploring the Sensitivity of LLMs' Decision-Making Capabilities: Insights from Prompt Variations and Hyperparameters"
_EMNLP/2023/Conference — EMNLP 2023 Findings_

### Official Review · Reviewer_nAFL · 2023-08-02

**Soundness:** 4

**Excitement:**

4: Strong: This paper deepens the understanding of some phenomenon or lowers the barriers to an existing research direction.

**Paper Topic And Main Contributions:**

The paper revisits the performance of large pretrained language models (LPLM) on a Horizon decision-making task studied by Binz and Schulz (2023). In the latter study it was found that there can be strong divergences between human and GPT3 performance on that task. The authors claim however that Binz and Schulz (2023) did not analyze the effects of prompt modification and temperature on the performance of LPLMs. The paper goes on to precisely study the change in performance of LPLM wrt. a Horizon decision-making task under prompt and temperature variation.

Their results empirically show that the decision-making capabilities of LPLM are indeed affected by both prompt and temperature. The former having a stronger effect than the latter. They also observe that, with adequate prompting, LPLM can display human-like exploration-exploitation trade-off.



**Questions For The Authors:**

(1) What is the temperature in the experiments shown in Fig 4; and why did you choose that value for this experiment? [I'd recommend adding this information to the paper]

(2) Can you comment on what should be a good strategy regarding the choice of temperature?

(3) is there a specific reason you did not study other tasks from Binz and Schulz (2023)?

(4) Why did you choose the Horizon decision-making task? i.e. instead of any of the other tasks in Binz and Schulz (2023)?

(5) is there a repo (even if it consists of simple scripts) with the code to reproduce the results (and download the datasets, etc)?

I understand that such a repo would require the users to have access to OpenAI APIs. But I also guess it could easily be adapted to e.g. Llama2?


**Reasons To Accept:**

The papers is concise, well written and makes a simple yet important point, namely that any study analyzing the characteristics, features and abilities of LPLM -- specially those studies coming out of the NLP community --- should consider the effect of prompting and hyperparameter variation on the LPLM performance.

I think it's a good short paper; and it complies nicely with the short paper definition of EMNLP.

**Reasons To Reject:**

As mentioned by the authors in the limitations section, their work only examines one of the tasks in Binz and Schulz (2023). It would have been interesting to see similar investigations wrt. the other tasks.



**Reproducibility:**

3: Could reproduce the results with some difficulty. The settings of parameters are underspecified or subjectively determined; the training/evaluation data are not widely available.

**Reviewer Confidence:**

3: Pretty sure, but there's a chance I missed something. Although I have a good feel for this area in general, I did not carefully check the paper's details, e.g., the math, experimental design, or novelty.

**Typos Grammar Style And Presentation Improvements:**

(1) Can you add (e.g. in the appendix) the equivalent of Fig 5 but for text-davinci-003 and text-davinci-002?

and, if the reason why you didn't add this figure is because the results are too similar, can you add a comment mentioning this?

(2) Can you add (e.g. in the appendix) the equivalent of Fig 4 but for the other two temperatures?

---

> ### Author Rebuttal · Authors · 2023-08-27
>
> > What is the temperature in the experiments shown in Fig 4; and why did you choose that value for this experiment?
>
> In Fig.4 the temperatures for gpt-2, gpt-3, gpt-3.5 are 1.0, 0.5, 1.0 respectively. We chose these temperatures because they showed the greatest learning effect (negative slope) in the horizon 6 task, based on Figure 2. We will add this information in the final version of the paper.
>
> > Can you comment on what should be a good strategy regarding the choice of temperature?
>
> It depends on how we want the model to perform. For example, in gpt-3.5 we found that when the temperature is set to 0 the model performs optimally. On the other hand, a temperature of 1.0 makes the model more human-like.
>
> > Is there a specific reason you did not study other tasks from Binz and Schulz (2023)?
>
> > Why did you choose the Horizon decision-making task? i.e. instead of any of the other tasks in Binz and Schulz (2023)?
>
> We found these results from the original paper to be the most worthy of further investigation because they involve the models’ ability to make economic decisions and implicitly calculate a regret function. These topics are under-explored in other work on LLM psychology.
>
> > Is there a repo (even if it consists of simple scripts) with the code to reproduce the results (and download the datasets, etc)?
>
> We have made the code available to reviewers as the supplementary material. Will make it available on Github upon publication.
> > I understand that such a repo would require the users to have access to OpenAI APIs. But I also guess it could easily be adapted to e.g. Llama2?
>
> Modifying the current code to carry out identical experiments on open-source models such as Llama 2 is straightforward, and we plan to incorporate comprehensive testing compatibility with open-source models in the final GitHub repository.

---

### Official Review · Reviewer_AUjq · 2023-08-05

**Soundness:** 4

**Excitement:**

4: Strong: This paper deepens the understanding of some phenomenon or lowers the barriers to an existing research direction.

**Paper Topic And Main Contributions:**

The Paper explores how different LLMs behave under different temperature and prompting strategy settings. They focus on the exploration-exploitation capabilities of these settings and compare it with human behavior.

**Questions For The Authors:**

A) Could you please explain why in Figure 2, the Horizon 1 and Horizon 6 tasks have a huge difference under "Trial 1"

**Reasons To Accept:**

Accepting it would allow the community to be more aware about considering the temperature and prompting strategies in their LLM research.

**Reasons To Reject:**

I dont see any reason to reject this paper

**Reproducibility:**

3: Could reproduce the results with some difficulty. The settings of parameters are underspecified or subjectively determined; the training/evaluation data are not widely available.

**Reviewer Confidence:**

5: Positive that my evaluation is correct. I read the paper very carefully and I am very familiar with related work.

---

> ### Author Rebuttal · Authors · 2023-08-27
>
> > Could you please explain why in Figure 2, the Horizon 1 and Horizon 6 tasks have a huge difference under "Trial 1"?
>
> In Figure 2, we believe the Horizon 1 and Horizon 6 tasks have a large difference under "Trial 1" because of the exploit-explore tradeoff. The prompt differs between the tasks (number of additional rounds left) and we'll make sure to show the differences in the appendix. In the Horizon 1 task, because the prompt explains there is only one trial, the LLM favors exploitation rather than exploration. In the Horizon 6 task, the prompt is as in Figure 1, so exploration is preferred.

---

### Official Review · Reviewer_57ap · 2023-08-09

**Soundness:** 2

**Excitement:**

3: Ambivalent: It has merits (e.g., it reports state-of-the-art results, the idea is nice), but there are key weaknesses (e.g., it describes incremental work), and it can significantly benefit from another round of revision. However, I won't object to accepting it if my co-reviewers champion it.

**Paper Topic And Main Contributions:**

This paper investigate on LLM's performance on the Horizon decision-making task, analyszing how LLMs respond to variations in prompts and hyperparameters.  LMs display a human-like exploration-exploitation tradeoff after simple adjustments to the prompt

The hyperparameters explored include:
temperature (0, 0.5, 1) and two variants of the Chain of Thought prompting technique (CoT and Quasi-CoT).


**Reasons To Accept:**

- psychologically guided prompt variants are interesting.
- comprehensive experiments

**Reasons To Reject:**

- Other factors in terms of prompt engineering can also play a role and hard to justify and do an exhaustive study on all variants. But more discussion on that will be helpful.

**Reproducibility:**

4: Could mostly reproduce the results, but there may be some variation because of sample variance or minor variations in their interpretation of the protocol or method.

**Reviewer Confidence:**

2: Willing to defend my evaluation, but it is fairly likely that I missed some details, didn't understand some central points, or can't be sure about the novelty of the work.

---

> ### Author Rebuttal · Authors · 2023-08-27
>
> > Other factors in terms of prompt engineering can also play a role and hard to justify and do an exhaustive study on all variants. But more discussion on that will be helpful.
>
> Our main motivation is to show that cognitive behavior varies according to prompt and temperature settings – indicating that tests related to these tasks should consider prompt and temperature before drawing conclusions. A more thorough study would certainly be worthwhile, but we believe the present contribution is enough for a short paper making it a focused contribution. We will make sure to investigate other factors that might influence the results.

---

### Official Review · Reviewer_zACN · 2023-08-13

**Soundness:** 2

**Excitement:**

2: Mediocre: This paper makes marginal contributions (vs non-contemporaneous work), so I would rather not see it in the conference.

**Paper Topic And Main Contributions:**

In this paper, the authors study the role hyperparameter and prompt tuning plays on the Horizon decision-making task–a pairwise comparison task in which agents operate under either “unequal information” or “equal information” settings–from Binz and Schulz (2023). The authors recreate the experiments from the paper and alter the original prompts and temperature of the tested LLMs. They find that models performance varies across different prompting and temperature setups.


**Reasons To Accept:**

The authors explore an interesting intersection of language modeling and human psychology.

**Reasons To Reject:**

Major weaknesses
* Poorly motivated direction: I am a bit unsure why minor alterations to temperature and prompting elucidate the “psychological behavior of LLMs” as the authors claim. I feel as though this research could have been more ambitious—simply demonstrating that the use of a slightly different prompting approach alters model outputs does not feel to be a substantive finding (especially given the numerous other research exploring model’s sensitivity to prompting strategies in more expansive domains and using much more robust methodology)
* The authors do not provide details necessary to judge the robustness of their experiments: number of trials, reasoning for temperatures chosen, reasoning for development of Quasi-CoT prompt
* Related works does not address multi-armed bandit setups which are critical for comprehension of methods
* Missing citations for key claims made; for example:
* 1) ln. 94-96: “In this task, humans show a characteristic trade-off of exploration and exploitation, favoring exploration in early trials and exploitation later…”
* 2) ln. 179-181: “...the identified limitations in LLMs, such as their inaccuracies in computing averages and sub-optimal exploration capabilities…”


Minor weaknesses
* Multiple grammatical errors and confusingly written at times
* Figure 3 is difficult to parse and contains major grammatical errors in the LM prompts
* The description of the Horizon task (described in Section 3) is difficult to parse. More precise language and motivation would really help better elucidate the Horizon task’s setup and utility.


The authors may want to consider reframing the work to include results for a different set of psychology benchmarks (ones that previous work has utilized in tandem use with LLMs) and demonstrate that conclusions vary as a function of the hyperparameters (i.e. showing the challenges with reproducibility when working with LLMs).

**Reproducibility:**

5: Could easily reproduce the results.

**Reviewer Confidence:**

4: Quite sure. I tried to check the important points carefully. It's unlikely, though conceivable, that I missed something that should affect my ratings.

**Typos Grammar Style And Presentation Improvements:**

Although there were many presentation errors/grammatical errors/typos; a few were:
* ln 29: “such” has no referent
* ln 58: missing “the” before “prompt”
* ln 57-63: very hard to parse


Figure 3, a few examples are:
* lesser → fewer
* summarization of output template is not a complete sentence
* choose → will choose
* computing sum → computing the sum

among many others…

---

> ### Author Rebuttal · Authors · 2023-08-27
>
> >Poorly motivated direction: I am a bit unsure why minor alterations to temperature and prompting elucidate the “psychological behavior of LLMs” as the authors claim. I feel as though this research could have been more ambitious—simply demonstrating that the use of a slightly different prompting approach alters model outputs does not feel to be a substantive finding (especially given the numerous other research exploring model’s sensitivity to prompting strategies in more expansive domains and using much more robust methodology)
>
> Regarding motivation, a more ambitious and extensive study would certainly be worthwhile, but we believe the present contribution is enough for a short paper making a focused contribution. Our motivation is to show that cognitive behavior varies according to prompt and temperature settings – indicating that tests related to these tasks should consider prompt and temperature before drawing conclusions.
>
> >The authors do not provide details necessary to judge the robustness of their experiments: number of trials, reasoning for temperatures chosen, reasoning for development of Quasi-CoT prompt.
>
> The number of trials is 6, following the setup from Binz and Schulz (2023) exactly. We will add this to the paper. The motivation for Quasi-CoT is to disentangle the effect of real CoT prompting from any effect of simply instructing the model to reason carefully.
>
>
> We will add further discussion and background on multiple bandits, basing our review on Sutton & Barto (2018).
>
> >Missing citations for key claims made; for example:
> ln. 94-96: “In this task, humans show a characteristic trade-off of exploration and exploitation, favoring exploration in early trials and exploitation later…”
> ln. 179-181: “...the identified limitations in LLMs, such as their inaccuracies in computing averages and sub-optimal exploration capabilities…”
>
> We will cite Wilson et al. (2014) for the claim on human psychology and Razeghi et al. (2022) for the lack of arithmetic accuracy in LLMs.
>
> We will address all the grammar and presentation issues mentioned.
>
> **References**
>
> 1. Sutton, Richard S., and Andrew G. Barto. Reinforcement learning: An introduction. MIT press, 2018.
> 2. Wilson RC, Geana A, White JM, Ludvig EA, Cohen JD. Humans use directed and random exploration to solve the explore-exploit dilemma. J Exp Psychol Gen. 2014 Dec;143(6):2074-81. doi: 10.1037/a0038199. Epub 2014 Oct 27. PMID: 25347535; PMCID: PMC5635655.
> 3. Razeghi, Yasaman, et al. "Impact of pretraining term frequencies on few-shot numerical reasoning." Findings of the Association for Computational Linguistics: EMNLP 2022. 2022.

---

### Meta-Review · Area_Chair_kKKH · 2023-09-29

**Recommendation:** 3

**Metareview:**

This work investigates how Large Language Models (LLMs) perform on a decision-making task that requires balancing exploration and exploitation. The authors show that LLMs’ behavior depends on the input prompts and the temperature settings, and that they can achieve human-like or superhuman results with simple adjustments. The authors suggest that LLM psychology studies should account for the variability of LLMs’ responses under different conditions.

Pros:
Demonstrated that LLMs can achieve superhuman performance in terms of minimizing regret on the Horizon task, which can have implications for designing more efficient and effective decision-making systems based on LLMs.

Cons:
Insufficient details and justifications for the experimental design and analysis

---

### Decision · Program_Chairs · 2023-10-07

**Decision:**

Accept-Findings

**Comment:**

This work investigates how Large Language Models (LLMs) perform on a decision-making task that requires balancing exploration and exploitation. The authors show that LLMs’ behavior depends on the input prompts and the temperature settings, and that they can achieve human-like or superhuman results with simple adjustments. The authors suggest that LLM psychology studies should account for the variability of LLMs’ responses under different conditions.

Pros:
Demonstrated that LLMs can achieve superhuman performance in terms of minimizing regret on the Horizon task, which can have implications for designing more efficient and effective decision-making systems based on LLMs.

Cons:
Insufficient details and justifications for the experimental design and analysis